# Semisynthetic Derivatives of Selected Amaryllidaceae Alkaloids as a New Class of Antimycobacterial Agents

**DOI:** 10.3390/molecules26196023

**Published:** 2021-10-04

**Authors:** Negar Maafi, Abdullah Al Mamun, Ondřej Janďourek, Jana Maříková, Kateřina Breiterová, Adéla Diepoltová, Klára Konečná, Anna Hošťálková, Daniela Hulcová, Jiří Kuneš, Eliška Kohelová, Darja Koutová, Marcela Šafratová, Lucie Nováková, Lucie Cahlíková

**Affiliations:** 1ADINACO Research Group, Department of Pharmaceutical Botany, Faculty of Pharmacy, Charles University, Heyrovskeho 1203, 500 05 Hradec Kralove, Czech Republic; negarm@faf.cuni.cz (N.M.); almamuna@faf.cuni.cz (A.A.M.); marikoj2@faf.cuni.cz (J.M.); breiterk@faf.cuni.cz (K.B.); HOSTA4AA@faf.cuni.cz (A.H.); hulcovd@faf.cuni.cz (D.H.); kohelove@faf.cuni.cz (E.K.); safratom@faf.cuni.cz (M.Š.); 2Department of Biological and Medical Sciences, Faculty of Pharmacy, Charles University, Heyrovskeho 1203, 500 05 Hradec Kralove, Czech Republic; JANDO6AA@faf.cuni.cz (O.J.); diepolta@faf.cuni.cz (A.D.); konecna@faf.cuni.cz (K.K.); 3Department of Bioorganic and Organic Chemistry, Faculty of Pharmacy, Charles University, Heyrovskeho 1203, 500 05 Hradec Kralove, Czech Republic; kunes@faf.cuni.cz; 4Department of Pharmacognosy, Faculty of Pharmacy, Charles University, Heyrovskeho 1203, 500 05 Hradec Kralove, Czech Republic; 5Department of Medical Biochemistry, Faculty of Medicine, Charles University, Simkova 870, 500 03 Hradec Kralove, Czech Republic; koutova.darja@lfhk.cuni.cz; 6Department of Analytical Chemistry, Faculty of Pharmacy, Charles University, Heyrovskeho 1203, 500 05 Hradec Kralove, Czech Republic; novakoval@faf.cuni.cz

**Keywords:** Amaryllidaceae, tuberculosis, galanthamine, 3-*O*-methylpancracine, analogues, antimycobacterial activity, cytotoxicity

## Abstract

The search for novel antimycobacterial drugs is a matter of urgency, since tuberculosis is still one of the top ten causes of death from a single infectious agent, killing more than 1.4 million people worldwide each year. Nine Amaryllidaceae alkaloids (AAs) of various structural types have been screened for their antimycobacterial activity. Unfortunately, all were considered inactive, and thus a pilot series of aromatic esters of galanthamine, 3-*O*-methylpancracine, vittatine and maritidine were synthesized to increase biological activity. The semisynthetic derivatives of AAs were screened for their in vitro antimycobacterial activity against *Mycobacterium tuberculosis* H37Ra and two other mycobacterial strains (*M. aurum*, *M. smegmatis*) using a modified Microplate Alamar Blue Assay. The most active compounds were also studied for their in vitro hepatotoxicity on the hepatocellular carcinoma cell line HepG2. In general, the derivatization of the original AAs was associated with a significant increase in antimycobacterial activity. Several pilot derivatives were identified as compounds with micromolar MICs against *M. tuberculosis* H37Ra. Two derivatives of galanthamine, **1i** and **1r**, were selected for further structure optimalization to increase the selectivity index.

## 1. Introduction

Despite the advances made in the treatment of tuberculosis (TB) in recent years, this disease still remains one of the main public health problems. TB is a devastating infectious disease caused by a tubercle bacillus, *Mycobacterium tuberculosis* complex. From this group of closely related species, the most common causative agent of TB is represented by *Mycobacterium*
*tuberculosis* (Mtb). The World Health Organization reports that about one-quarter of the global population has latent TB, and around 10.0 million people developed TB globally in 2019, with an estimated 1.4 million TB deaths among HIV-negative people, including 208,000 deaths among HIV-positive individuals [1]. Current treatment for drug-sensitive TB consists of standard first-line drugs, including isoniazid, rifampicin, ethambutol and pyrazinamide, used for at least six months [2]. However, the increasing prevalence of multi-drug-resistant (MDR) strains of Mtb, with resistance to at least rifampicin and isoniazid, has rendered these first-line antibiotics significantly less effective. Therefore, the search for new effective anti-tuberculosis (anti-TB) drugs, possessing activities against various subpopulations of Mtb, or with novel modes of action, remains a serious challenge for medicinal chemistry.

The poor efficiency of identifying new anti-TB and antibacterial drugs by the screening of pharmaceutical libraries is associated with limited chemical diversity within these collections. Additionally, most anti-TB and antibacterial drugs in general do not follow Lipinski´s rule of 5, which defines the optimal drug-like features, whereas pharmaceutical compounds are biased towards these properties [3]. 

Thus, the key question is: How do we search for new anti-TB and antibacterial drugs and where do we look for them?

One of the options represents different types of natural products (NPs). The importance of natural products (NPs) in drug discovery has been extensively documented, including their contribution to the development of present drugs [4,5]. The chemical diversity of NPs is more closely aligned with drugs than synthetic libraries, thus making them ideal candidates for drug-discovery projects. Moreover, naturally occurring pure compounds from higher and lower forms of plants as well as microorganisms and marine organisms have shown antimycobacterial potential in different studies. Without a doubt, the most studied group of natural products are alkaloids, which are classified into various structural types according to their biosynthetic origin. Recently, several reviews or perspective pieces have been published on this topic [6,7,8,9].

The plant family Amaryllidaceae is known for its structurally unique isoquinoline alkaloids called Amaryllidaceae alkaloids (AAs). Traditionally, plants of this family have been used as folk herbal remedies against various diseases by indigenous people around the world [10]. The biological activity of the plants is connected to the presence of AAs, which demonstrate a wide range of biological activities including antitumor, antibacterial, antioxidant, antiparasitic, antifungal, anti-inflammatory and insect antifeedant effects, as well as acetylcholin- and butyrylcholinesterase inhibitory activities [11,12,13]. The most known compound within the AAs is galanthamine, which is approved for the treatment of Alzheimer´s disease under the commercial names Razadyne^©^ in the USA and Reminyl^©^ in Europe and other countries [14]. Other interesting AAs are lycorine, pancratistatine and haemanthamine, which are intensively studied for their antitumor properties [15,16,17,18].

Given the fact that AAs have been, and are being, intensively studied for a wide range of biological activities, it is surprising that these natural products have not yet been addressed in terms of their antimycobacterial activity. Due to the fact that a number of AAs can be isolated from plants in large quantities, they are also an interesting target for the preparation of semisynthetic derivatives and detailed structure–activity relationship (SAR) studies leading to the optimalization of the structure.

As a part of our ongoing research on AAs and their semisynthetic derivatives as potential drugs, this work reports the antimycobacterial activity of selected AAs and their pilot semisynthetic derivatives. 

## 2. Results and Discussion

### 2.1. Synthesis of Galanthamine Derivatives (***1a***–***1s***)

Since galanthamine is used as a drug for the treatment of Alzheimer´s disease (AD), the original goal of this study was to develop new galanthamine derivatives, and investigate them as novel chemical entities with improved cholinesterases inhibition profiles. Unfortunately, derivatization of the hydroxy group at position C-6 with differently substituted aromatic moieties was associated with the loss of biological activity. Thus, selected galanthamine derivatives were randomly screened for their antimycobacterial activity. Surprisingly, a few of them demonstrated promising antimycobacterial activity, and thus the remaining derivatives were also screened.

The structural aspects of galanthamine (**1**) allowed us to prepare novel chemical entities by derivatization of the free hydroxyl group at position C-6 and to inspect the structure–activity relationship of the novel derivatives. Chemical modification of galanthamine was feasible because of its previous large-scale isolation from *Narcissus pseudonarcissus* cv. Carlton. Modification of the structure was inspired by previous structural changes of haemanthamine, ambelline and vittatine [12,19,20,21]. Using the same method described in our previous reports, the hydroxy group at C-6 was acylated with differently substituted benzoyl chlorides, 1- and 2-naphthoyl chlorides and 2-furoyl chloride affording the corresponding esters **1a**–**1s** (Figure 1). Derivatives (except **1g**–**1i**) were converted to their hydrochlorides to increase their solubility. The structures of the newly synthesized compounds were determined by MS, HRMS and 1D- and 2D-NMR spectroscopic techniques (see Appendix A). The data were fully consistent with the proposed structures. The yield of all reactions was >52% (the yield for each reaction is specified in the experimental part).

### 2.2. Synthesis of 3-O-methylpancracine (***3a***–***3g***), Vittatine (***4a***, ***4b***) and Maritidine Derivatives (***5a***, ***5b***) 

Derivatives of 3-*O*-methylpancracine were originally synthesized to study the cytotoxic potential of montanine-type Amaryllidaceae alkaloids. The promising antiproliferative properties of some representatives of this structural type of AA, and the limited presence of these alkaloids in plant material, initiated the total synthesis and production of synthetic analogues of these alkaloids [22,23]. Among these, the montanine-type structure was successfully achieved with intramolecular rearrangement of haemanthamine, which is considered a convenient semi-synthetic method for achieving the montanine-type skeleton due to the ready availability of haemanthamine from plants of the Amaryllidaceae family [23]. Unfortunately, similar to galanthamine derivatives, the structural modification of the montanine skeleton was associated with the loss of cytotoxic activity, and thus we decided to screen selected derivatives of 3-*O*-methylpancracine for antimycobacterial activity. 

Using a skeletal rearrangement of haemanthamine (**2**) and subsequent acylation, different substituted esters of 3-*O*-methylpancracine (**3**) were synthesized. The structural modifications of **3** are displayed in Figure 2.

To study the role of the structural type of AAs in the structure of the tested esters, pilot derivatives of vittatine (**4**) and maritidine (**5**), corresponding to the active galanthamine analogues, were also synthesized for more detailed SAR studies (Figure 3). The amount of both alkaloids isolated was limited and allowed the preparation of only two pilot derivatives from each alkaloid (**4a**–**4b**; **5a**–**5b**). The structural modifications of **4** and **5** are displayed in Figure 3. The structures were determined by HRMS and 1D- and 2D-NMRspectroscopic techniques (see Appendix A). The yield of all reactions was >42% (the yield for each reaction is specified in the experimental part). All synthesized derivatives were converted to their hydrochlorides to improve the solubility for antimycobacterial assay.

### 2.3. Antimycobacterial Activity of Selected Amaryllidaceae Alkaloids (***1***–***9***) and Their Synthesized Derivatives (***1a***–***1s***, ***3a***–***3g***, ***4a***–***4b*** and ***5a***–***5b***)

In the current study, nine AAs of various structural types isolated previously from different Amaryllidaceae plants (except 3-*O*-methylpancracine, which has been synthesized within the current study; Figure 4) [19,20,24], along with 30 newly synthesized derivatives of galanthamine (**1a**–**1s**), 3-*O*-methylpancracine (**3a**–**3g**), vittatine (**4a**–**4b**) and maritidine (**5a**–**5b**) have been screened for their antimycobacterial activity against three different mycobacterial strains: *M. tuberculosis* H37Ra, *M. aurum* and *M. smegmatis*. The last two are fast-growing strains that do not cause tuberculosis in humans, but are much safer to handle. Their phenotype and genotype are close to those of Mtb, which allows their use as a surrogate model of Mtb [25]. Antimycobacterial activity is expressed as the minimum inhibitory concentration (MIC).

Although alkaloids of different structural types and biosynthetic origin have been reported as compounds with promising antimycobacterial activities, all screened AAs lacked any antimycobacterial activity (Table 1). These results where surprising to us, especially in view of the fact that these natural products demonstrate a wide range of biological activities. 

As already mentioned above, the analogues of galanthamine were originally intended to study their cholinesterases inhibition potency, but all derivatives were considered inactive. Thus, selected derivatives were initially screened for antimycobacterial activity, and, surprisingly, some of them showed strong activity, and thus the remaining derivatives were also tested. Interestingly, derivatization of the C-6 hydroxy group was in all cases associated with a significant increase in antimycobacterial activity against all studied *Mycobacterium* strains (MIC = 1.56–62.5 µg/mL), with a few exceptions (Table 1). Among the substituted benzoyl-derivatives of galanthamine, the strongest activity was related to the presence of either an isobutyl- (**1h**) or butyl-chain (**1i**) in the *para* position on the benzene ring. Both derivatives displayed impressive activity against all the studied strains (MICs = 3.125–7.81 µg/mL and MICs = 1.56–7.81 µg/mL, respectively). The most sensitive strain was *M. tuberculosis* H37Ra with an MIC = 3.125 µg/mL (6.9 µM) for 6-*O*-(4-*tert-*butylbenzoyl)galanthamine (**1h**) and MIC = 1.56 µg/mL (3.5 µM) for 6-*O*-(4-butylbenzoyl)galanthamine (**1i**). The presence of a longer chain on the aromatic ring seems to be responsible for a dramatic increase in antimycobacterial activity. These findings can be deduced by comparing the activities of derivatives **1d** and **1i**, which differ only in the size of the substituent on the aromatic ring in the *para* position. When there is a methyl group, as in 6-*O*-(4-methylbenzoyl)galanthamine (**1d**), the analogue shows up to ten times lower activity than the substance with a butyl-substitution, as in 6-*O*-(4-butylbenzoyl)galanthamine (**1i**) (Table 1). The presence of two or three substituents (methyl-, methoxy- or ethoxy- group) on the benzene ring (like in **1e**, **1f**, **1g** and **1p**) was also associated with an interesting increase in antimycobacterial activity. Within these compounds, 6-*O*-(3,5-diethoxybenzoyl)galanthamine (**1p**) showed promising antimycobacterial activity, with MICs = 3.91–15.625 µg/mL (7.6–30.2 µM). Interesting activity was also demonstrated by the naphthoyl-derivatives of galanthamine (**1q** and **1r**). Among these, 6-*O*-(2-naphthoyl)galanthamine (**1r**) showed up to three times stronger activity, with MICs = 1.98–7.81 µg/mL, in comparison with the galanthamine analogue, for which the 1-naphthoyl chloride was used for esterification (**1q**, MICs = 6.25–7.81 µg/mL).

These findings encouraged us to study the antimycobacterial activity of further Amaryllidaceae alkaloids (previously isolated or synthesized in our laboratory) with the same derivatization as the most active galanthamine analogues. Thus, the pilot series was tested on analogues of 3-*O*-methylpancracine (**3a**–**3g**), corresponding to the most active galanthamine-derivatives. In general, the derivatives of 3-*O*-methylpancracine demonstrated either comparable or slightly lower/higher antimycobacterial activity. Interestingly, among the naphthoyl derivatives of **3**, the stronger antimycobacterial potency was associated with the presence of a 1-naphthoyl-substituent (**3f,** MICs = 3.91 µg/mL for all studied strains), instead of a 2-naphthoyl-substituent, like in the galanthamine analogue (**1r**). To study this phenomenon, two naphthoyl-analogues of vittatine (**4**) were synthesized and tested (**4a**–**4b**). In this case, the same antimycobacterial activity was obtained for both analogues (MICs = 3.91–7.81 µg/mL). The limited amount of maritidine (**5**) isolated allowed preparation of only two derivatives (**5a**, **5b**), both of which also showed interesting activity (MICs = 3.91–15.625 µg/mL; Table 1).

### 2.4. Lipophilicity Versus Activity

Lipophilicity is one of the most important physicochemical properties of a compound, which seems to be an important factor related to cell transmembrane transport. The mycobacterial cell wall is rich in mycolic acids, which efficiently prevent the penetration of drugs. Drugs with higher lipophilicity may exert better activity against *M. tuberculosis* [26], and thus we calculated logP/ClogP values using the ChemBioDraw Ultra program (ver 18.1). The ClogP value is correlated directly to the molecular hydrophobicity, and thereby, to the diffusion through the biological membranes, i.e., through the mycobacterial cell wall. The low antimycobacterial activity of the tested alkaloids is in good agreement with their values for log*P*/ClogP (logP ≤ 1.45; ClogP ≤ 1.08, Table 1). Actually, the novel active compounds in this study (MIC ≤ 7.81 µg/mL) reached logP values ranging between 3.72 and 5.27 and ClogP 4.20–5.99 (ClogP), indicating their potential for development as antimycobacterial drugs.

### 2.5. Cytotoxicity of Selected Derivatives of Amaryllidaceae Alkaloids

The most active derivatives were further evaluated for in vitro cytotoxicity on hepatocellular carcinoma cells (HepG2) using an MTT assay. The HepG2 cell line serves as an in vitro model for hepatotoxicity for early drug screening. Moreover, the hepatocellular model was chosen since antitubercular drugs are known to carry the risk of hepatotoxicity and represent the most likely target for chronic toxicity of antitubercular drugs, which often complicates the six-months-long therapy for TB [27]. 

The evaluation of cytotoxicity allows the calculation of selectivity indexes (SI) as the ratio of IC_50,HepG2_ to MIC of *Mtb* H37Ra (Table 1). In general, values of SI higher than 10 indicate more acceptable toxicity (analogous to the therapeutic index). The active compounds were screened at an initial concentration of 50 µM, and the IC_50_ values were subsequently determined. The effective cytotoxic concentrations of all tested compounds are listed in Table 1. Among these compounds, 6-*O*-(2,4,6-trimethylbenzoyl)galanthamine (**1g**) exhibited the highest cytotoxicity with an IC_50_ value = 13.9 ± 0.8 µM. The most active derivatives in the antimycobacterial assay (**1i** and **1r)** showed cytotoxicity with IC_50_ values of 14.7 ± 1.6 µM, and 21.2 ± 3.8 µM, reaching an SI of 4.20 for **1i** and 5.17 for **1r**, which indicates the potential risk of hepatotoxicity. In subsequent structure optimalization of the active antitubercular derivatives, the SI must be increased to reduce the hepatotoxicity of the developed compounds.

## 3. Materials and Methods

### 3.1. General Experimental Procedures

All solvents were treated using standard techniques before use. All reagents and catalysts were purchased from Sigma Aldrich, Czech Republic, and used without purification. NMR spectra were recorded in CDCl_3_ on either a VNMR S500 (Varian) spectrometer operating at 500 MHz for ^1^H and 125.7 MHz for ^13^C or a Jeol JNM-ECZ600R operating at 600 MHz for ^1^H and 151 MHz for ^13^C at an ambient temperature. The residual signal of CHCl_3_ (*δ* 7.26 ppm) was a reference for ^1^H NMR spectra, and the central signal of the CDCl_3_ signals (*δ* 77.0 ppm) was used as a reference for proton-decoupled ^13^C NMR spectra. The coupling constant (*J*) is given in Hz, and the chemical shifts are reported in ppm. For unambiguous assignment of ^1^H and ^13^C NMR signals, ESI-HRMS were obtained with a Waters Synapt G2-Si hybrid mass analyzer of a quadrupole-time-of-flight (Q-TOF) type, coupled to a Waters Acquity I-Class UHPLC system. EI-MS were obtained on an Agilent 7890A GC 5975 inert MSD operating in EI mode at 70 eV (Agilent Technologies, Santa Clara, CA, USA). A DB-5 column (30 m × 0.25 mm × 0.25 μm, Agilent Technologies, USA) was used with a temperature program: 100–180 °C at 15°C/min, 1 min hold at 180 °C, and 180–300 °C at 5 °C/min and 5 min hold at 300 °C; and a detection range of *m/z* 40–600. The injector temperature was 280 °C. The flow-rate of the carrier gas (helium) was 0.8 mL/min. A split ratio of 1:15 was used. UV and ECD spectra were recorded on a JASCO J-815 CD spectrometer. Compounds on TLC plates were observed under UV light (254 and 366 nm) and visualized by spraying with Dragendorff’s reagent. 

### 3.2. Amaryllidaceae Alkaloids

Alkaloids **1** and **9** were isolated from *Narcissus pseudonarcissus* cv. Carlton, alkaloids **2** and **5–7** from *Zephyranthes citrina* and alkaloids **4** and **8** from *Hippeastrum x hybridum* cv. Ferrari, as previously reported in our studies [12,20,28]. The purity of all compounds, verified by NMR, was 95%.

### 3.3. Preparation of Galanthamine (***1***), Vittatine (***4***) and Maritidine (***5***) Derivatives

The same procedure as described previously was used to afford the corresponding esters **1a**–**1s**, **4a**–**4b** and **5a**–**5b** [20,29].

#### 3.3.1. 6-*O*-benzoylgalanthamine (**1a**) 

Yield 64 mg (96%); white amorphous solid; $\left[\alpha \right]{\;}_{D}^{28}$= −25.3° (c 0.19, MeOH); ^1^H NMR (500 MHz, CDCl_3_) δ: 8.08–8.04 (m, 2H), 7.55–7.47 (m, 1H), 7.41–7.34 (m, 2H), 6.69 (d, *J* = 8.1 Hz, 1H), 6.59 (d, *J* = 8.1 Hz, 1H), 6.37 (d, *J* = 10.3 Hz, 1H), 6.05 (dd, *J* = 10.3 Hz, *J* = 5.2 Hz, 1H), 5.63–5.56 (m, 1H), 4.69–4.65 (m, 1H), 4.16 (d, *J* = 15.2 Hz, 1H), 3.90 (s, 3H), 3.69 (d, *J* = 15.2 Hz, 1H), 3.36 (t, *J* = 13.8 Hz, 1H), 3.14–3.05 (m, 1H), 2.84–2.75 (m, 1H), 2.42 (s, 3H), 2.25–2.12 (m, 2H), 1.68–1.60 (m, 1H) ^13^C NMR (126 MHz, CDCl_3_) δ: 166.2, 146.7, 144.0, 132.7, 132.0, 131.0, 130.6, 129.9, 129.3, 128.1, 122.8, 121.2, 111.4, 86.3, 63.6, 60.3, 56.0, 53.7, 48.2, 41.7, 34.2, 28.0; ESI-HRMS *m/z* calcd for C_24_H_26_NO_4_ [M + H]^+^ 392.1856 found 392.1857.

#### 3.3.2. 6-*O*-(2-methylbenzoyl)galanthamine (**1b**)

Yield 60 mg (85%); white amorphous solid; $\left[\alpha \right]{\;}_{D}^{28}$= −13.3° (c 0.12, MeOH); ^1^H NMR (500 MHz, CDCl_3_) δ: 8.03 (dd, *J* = 7.9 Hz, *J* = 1.5 Hz, 1H), 7.35 (td, *J* = 7.9 Hz, *J* = 1.5 Hz, 1H), 7.27–7.14 (m, 2H), 6.68 (d, *J* = 8.1 Hz, 1H), 6.59 (d, *J* = 8.1 Hz, 1H), 6.35 (d, *J* = 10.3 Hz, 1H), 6.04 (dd, *J* = 10.3 Hz, *J* = 5.1 Hz, 1H), 5.60–5.53 (m, 1H), 4.65–4.63 (m, 1H), 4.15 (d, *J* = 15.2 Hz, 1H), 3.87 (s, 3H), 3.70 (d, *J* = 15.2 Hz, 1H), 3.35 (t, *J* = 12.8 Hz, 1H), 3.15–3.05 (m, 1H), 2.85–2.78 (m, 1H), 2.59 (s, 3H), 2.42 (s, 3H), 2.24–2.13 (m, 2H), 1.69–1.60 (m, 1H); ^13^C NMR (126 MHz, CDCl_3_) δ: 167.2, 146.7, 144.0, 140.4, 132.0, 131.8, 131.5, 131.4, 130.9, 129.7, 129.2, 125.6, 122.9, 121.3, 111.5, 86.3, 63.4, 60.4, 56.0, 53.8, 48.1, 41.8, 34.3, 27.9, 21.7; ESI-HRMS *m/z* calcd for C_25_H_28_NO_4_ [M + H]^+^ 406.2013 found 406.2013.

#### 3.3.3. 6-*O*-(3-methylbenzoyl)galanthamine (**1c**)

Yield 41 mg (58%); white amorphous solid; $\left[\alpha \right]{\;}_{D}^{28}$= −17.0° (c 0.16, MeOH); ^1^H NMR (500 MHz, CDCl_3_) δ: 7.89–7.84 (m, 2H), 7.34–7.30 (m, 1H), 7.29–7.23 (m, 1H), 6.69 (d, *J* = 8.1 Hz, 1H), 6.59 (d, *J* = 8.1 Hz, 1H), 6.36 (d, *J* = 10.3 Hz, 1H), 6.03 (dd, *J* = 10.3 Hz, *J* = 5.2 Hz, 1H), 5.62–5.57 (m, 1H), 4.68–4.63 (m, 1H), 4.14 (d, *J* = 15.2 Hz, 1H), 3.88 (s, 3H), 3.68 (d, *J* = 15.2 Hz, 1H), 3.34 (t, *J* = 13.7 Hz, 1H), 3.12–3.03 (m, 1H), 2.81–2.73 (m, 1H), 2.41 (s, 3H), 2.37 (s, 3H), 2.23–2.11 (m, 2H), 1.66–1.59 (m, 1H); ^13^C NMR (126 MHz, CDCl_3_) δ: 166.3, 146.7, 144.0, 137.8, 133.4, 132.0, 130.9, 130.4, 130.3, 129.3, 127.9, 127.0, 122.8, 121.2, 111.4, 86.3, 63.4, 60.3, 55.9, 53.7, 48.1, 41.8, 34.2, 27.9, 21.1; ESI-HRMS *m/z* calcd for C_25_H_28_NO_4_ [M + H]^+^ 406.2013 found 406.2015.

#### 3.3.4. 6-*O*-(4-methylbenzoyl)galanthamine (**1d**)

Yield 38 mg (51%); white amorphous solid; $\left[\alpha \right]{\;}_{D}^{28}$= −27.4° (c 0.35, MeOH); ^1^H NMR (500 MHz, CDCl_3_) δ: 7.98–7.90 (m, 2H, AA′BB′), 7.20–7.14 (m, 2H, AA′BB′), 6.68 (d, *J* = 8.1 Hz, 1H), 6.58 (d, *J* = 8.1 Hz, 1H), 6.35 (d, *J* = 10.3 Hz, 1H), 6.04 (dd, *J* = 10.3 Hz, *J* = 5.2 Hz, 1H), 5.60–5.54 (m, 1H), 4.68–4.62 (m, 1H), 4.14 (d, *J* = 15.1 Hz, 1H), 3.89 (s, 3H), 3.68 (d, *J* = 15.1 Hz, 1H), 3.34 (t, *J* = 12.8 Hz, 1H), 3.12–3.03 (m, 1H), 2.82–2.73 (m, 1H), 2.41 (s, 3H), 2.38 (s, 3H), 2.23–2.11 (m, 2H), 1.66–1.55 (m, 1H); ^13^C NMR (126 MHz, CDCl_3_) δ: 166.2, 146.7, 143.9, 143.2, 132.0, 130.8, 129.9, 129.3, 128.8, 127.8, 122.9, 121.1, 111.5, 86.3, 63.4, 60.3, 56.0, 53.7, 48.1, 41.7, 34.2, 28.0, 21.6; ESI-HRMS *m/z* calcd for C_25_H_28_NO_4_ [M + H]^+^ 406.2013 found 406.2019.

#### 3.3.5. 6-*O*-(2,3-dimethylbenzoyl)galanthamine (**1e**)

Yield 45 mg (61%); white amorphous solid; $\left[\alpha \right]{\;}_{D}^{28}$= −21.4° (c 0.17, MeOH); ^1^H NMR (600 MHz, CDCl_3_) δ: 7.74 (d, *J* = 8.0 Hz, 1H), 7.22 (d, *J* = 8.0 Hz, 1H), 7.04 (t, *J* = 8.0 Hz, 1H), 6.66 (d, *J* = 8.1 Hz, 1H), 6.57 (d, *J* = 8.1 Hz, 1H), 6.33 (d, *J* = 10.3 Hz, 1H), 6.02 (dd, *J* = 10.3 Hz, *J* = 5.2 Hz, 1H), 5.55–5.50 (m, 1H), 4.64–4.60 (m, 1H), 4.14 (d, *J* = 15.2 Hz, 1H), 3.84 (s, 3H), 3.68 (d, *J* = 15.2 Hz, 1H), 3.34 (t, *J* = 13.5 Hz, 1H), 3.10–3.05 (m, 1H), 2.81–2.75 (m, 1H), 2.43 (s, 3H), 2.40 (s, 3H), 2.27 (s, 3H), 2.20–2.11 (m, 2H), 1.66–1.60 (m, 1H); ^13^C NMR (151 MHz, CDCl_3_) δ: 168.2, 146.7, 144.1, 137.9, 137.5, 133.0, 132.0, 130.9, 130.9, 128.8, 125.0, 123.0, 121.3, 111.5, 86.3, 63.6, 60.4, 56.0, 53.7, 48.1, 41.7, 34.2, 27.9, 20.5, 16.5; ESI-HRMS *m/z* calcd for C_26_H_30_NO_4_ [M + H]^+^ 420.2169 found 420.2179.

#### 3.3.6. 6-*O*-(3,5-dimethylbenzoyl)galanthamine (**1f**)

Yield 41 mg (56%); white amorphous solid; $\left[\alpha \right]{\;}_{D}^{28}$= −15.3° (c 0.23, MeOH); ^1^H NMR (500 MHz, CDCl_3_) δ: 7.67 (s, 2H), 7.14 (s, 1H), 6.69 (d, *J* = 8.1 Hz, 1H), 6.59 (d, *J* = 8.1 Hz, 1H), 6.35 (d, *J* = 10.3 Hz, 1H), 6.02 (dd, *J* = 10.3 Hz, *J* = 5.1 Hz, 1H), 5.64–5.52 (m, 1H), 4.67–4.62 (m, 1H), 4.14 (d, *J* = 15.1 Hz, 1H), 3.87 (s, 3H), 3.69 (d, *J* = 15.1 Hz, 1H), 3.34 (t, *J* = 13.6 Hz, 1H), 3.12–3.00 (m, 1H), 2.82–2.70 (m, 1H), 2.41 (s, 3H), 2.33 (s, 6H), 2.22–2.06 (m, 2H), 1.66–1.59 (m, 1H); ^13^C NMR (126 MHz, CDCl_3_) δ: 166.5, 146.7, 144.0, 137.6, 134.3, 132.0, 130.7, 130.3, 129.2, 127.5, 122.8, 121.2, 111.4, 86.3, 63.2, 60.3, 55.9, 53.7, 48.0, 41.7, 34.2, 27.9, 21.0; ESI-HRMS *m/z* calcd for C_26_H_30_NO_4_ [M + H]^+^ 420.2169 found 420.2174.

#### 3.3.7. 6-*O*-(2,4,6-trimethylbenzoyl)galanthamine (**1g**)

Yield 50 mg (66%); yellow amorphous solid; $\left[\alpha \right]{\;}_{D}^{28}$= −12.6° (c 0.19, MeOH); ^1^H NMR (600 MHz, CDCl_3_) δ: 6.77 (s, 2H), 6.61 (d, *J* = 8.1 Hz, 1H), 6.54 (d, *J* = 8.1 Hz, 1H), 6.28 (d, *J* = 10.3 Hz, 1H), 6.02 (dd, *J* = 10.3 Hz, *J* = 4.9 Hz, 1H), 5.57–5.48 (m, 1H), 4.61–4.53 (m, 1H), 4.10 (d, *J* = 15.1 Hz, 1H), 3.72 (s, 3H), 3.66 (d, *J* = 15.1 Hz, 1H), 3.29 (t, *J* = 13.9 Hz, 1H), 3.11–2.99 (m, 1H), 2.83–2.75 (m, 1H), 2.38 (s, 3H), 2.25 (s, 6H), 2.23 (s, 3H), 2.21–2.12 (m, 2H), 1.64–1.58 (m, 1H); ^13^C NMR (151 MHz, CDCl_3_) δ: 169.9, 146.8, 144.0, 139.0, 135.6, 131.9, 130.9, 130.6, 129.0, 128.3, 122.7, 121.3, 112.0, 85.9, 64.2, 60.4, 56.1, 53.7, 48.0, 41.9, 34.4, 28.0, 21.1, 19.8. ESI-HRMS *m/z* calcd for C_27_H_32_NO_4_ [M + H]^+^ 434.2326 found 434.2332.

#### 3.3.8. 6-*O*-(4-tert-butylbenzoyl)galanthamine (**1h**)

Yield 70 mg (89%); yellow amorphous solid; $\left[\alpha \right]{\;}_{D}^{28}$= −25.5° (c 0.44, MeOH); ^1^H NMR (600 MHz, CDCl_3_) δ: 7.98–7.93 (m, 2H, AA′BB′), 7.38– 7.34 (m, 2H, AA′BB′), 6.67 (d, *J* = 8.1 Hz, 1H), 6.57 (d, *J* = 8.1 Hz, 1H), 6.33 (d, *J* = 10.2 Hz, 1H), 6.02 (dd, *J* = 10.2 Hz, *J* = 5.2 Hz, 1H), 5.57–5.52 (m, 1H), 4.65–4.62 (m, 1H), 4.14 (d, *J* = 15.1 Hz, 1H), 3.88 (s, 3H), 3.67 (d, *J* = 15.1 Hz, 1H), 3.34 (t, *J* = 13.3 Hz, 1H), 3.11–3.03 (m, 1H), 2.78–2.70 (m, 1H), 2.39 (s, 3H), 2.21–2.10 (m, 2H), 1.64–1.58 (m, 1H), 1.30 (s, 9H); ^13^C NMR (151 MHz, CDCl_3_) δ: 166.3, 156.2, 146.8, 144.0, 132.1, 130.9 129.8, 129.2, 127.8, 125.0, 123.0, 121.2, 111.5, 86.3, 63.4, 60.3, 56.0, 53.7, 48.2, 41.6, 34.9, 34.2, 31.1, 28.0; ESI-HRMS *m/z* calcd for C_28_H_34_NO_4_ [M + H]^+^ 448.2482 found 448.2494.

#### 3.3.9. 6-*O*-(4-butylbenzoyl)galanthamine (**1i**)

Yield 75 mg (96%); yellow amorphous solid; $\left[\alpha \right]{\;}_{D}^{28}$= −20.4° (c 0.45, MeOH); ^1^H NMR (600 MHz, CDCl_3_) δ: 7.96–7.92 (m, 2H, AA′BB′), 7.17–7.12 (m, 2H, AA′BB′) , 6.67 (d, *J* = 8.1 Hz, 1H), 6.57 (d, *J* = 8.1 Hz, 1H), 6.33 (d, *J* = 10.3 Hz, 1H), 6.02 (dd, *J* = 10.3 Hz, *J* = 5.2 Hz, 1H), 5.58–5.52 (m, 1H), 4.68–4.60 (m, 1H), 4.13 (d, *J* = 15.1 Hz, 1H), 3.88 (s, 3H), 3.67 (d, *J* = 15.1 Hz, 1H), 3.33 (t, *J* = 13.8 Hz, 1H), 3.09–3.03 (m, 1H), 2.78–2.73 (m, 1H), 2.62 (t, *J* = 7.4 Hz, 2H), 2.39 (s, 3H), 2.20–2.10 (m, 2H), 1.66–1.51 (m, 3H), 1.32 (dq, *J* = 14.8 Hz, *J* = 7.4 Hz, 2H), 0.90 (t, *J* = 7.4 Hz, 3H); ^13^C NMR (151 MHz, CDCl_3_) δ: 166.3, 148.2, 146.8, 144.0, 132.1, 130.9, 130.0, 129.3, 128.2, 128.0, 123.0, 121.2, 111.5, 86.3, 63.4, 60.3, 56.1, 53.7, 48.2, 41.7, 35.7, 34.2, 33.3, 28.0, 22.2, 13.8; ESI-HRMS *m/z* calcd for C_28_H_34_NO_4_ [M + H]^+^ 448.2482 found 448.2493.

#### 3.3.10. 6-*O*-(2-methoxybenzoyl)galanthamine (**1j**)

Yield 58 mg (81%); white amorphous solid; $\left[\alpha \right]{\;}_{D}^{28}$= −26.0° (c 0.20, MeOH); ^1^H NMR (600 MHz, CDCl_3_) δ: 7.93 (dd, *J* = 7.7 Hz, *J* = 1.9 Hz, 1H), 7.46–7.36 (m, 1H), 6.97–6.83 (m, 2H), 6.65 (d, *J* = 8.0 Hz, 1H), 6.57 (d, *J* = 8.0 Hz, 1H), 6.32 (d, *J* = 10.1 Hz, 1H), 6.01 (dd, *J* = 10.1 Hz, *J* = 5.3 Hz, 1H), 5.57–5.53 (m, 1H), 4.66–4.58 (m, 1H), 4.13 (d, *J* = 15.1 Hz, 1H), 3.87 (s, 3H), 3.85 (s, 3H), 3.67 (d, *J* = 15.1 Hz, 1H), 3.33 (t, *J* = 13.8 Hz, 1H), 3.10–3.04 (m, 1H), 2.82–2.72 (m, 1H), 2.40 (s, 3H), 2.23–2.08 (m, 2H), 1.65–1.56 (m, 1H); ^13^C NMR (151 MHz, CDCl_3_) δ: 165.1, 159.6, 146.7, 144.0, 133.4, 132.7, 132.1, 130.8, 129.0, 123.1, 121.3, 120.0, 119.9, 111.7, 111.5, 86.4, 63.3, 60.3, 56.0, 55.9, 53.7, 48.1, 41.6, 34.2, 27.9; ESI-HRMS *m/z* calcd for C_25_H_28_NO_5_ [M + H]^+^ 422.1962 found 422.1970.

#### 3.3.11. 6-*O*-(3-methoxybenzoyl)galanthamine (**1k**)

Yield 64 mg (90%); white amorphous solid; $\left[\alpha \right]{\;}_{D}^{28}$= −19.4° (c 0.35, MeOH); ^1^H NMR (500 MHz, CDCl_3_) δ: 7.67 (d, *J* = 7.6 Hz, 1H), 7.59–7.55 (m, 1H), 7.28 (t, *J* = 7.6 Hz, 1H), 7.06 (dd, *J* = 7.6 Hz, *J* = 2.7 Hz, 1H), 6.69 (d, *J* = 8.1 Hz, 1H), 6.60 (d, *J* = 8.1 Hz, 1H), 6.35 (d, *J* = 10.3 Hz, 1H), 6.05 (dd, *J* = 10.3 Hz, *J* = 5.1 Hz, 1H), 5.65–5.55 (m, 1H), 4.68–4.63 (m, 1H), 4.18 (d, *J* = 15.1 Hz, 1H), 3.86 (s, 3H), 3.84 (s, 3H), 3.72 (d, *J* = 15.1 Hz, 1H), 3.38 (t, *J* = 13.6 Hz, 1H), 3.17–3.05 (m, 1H), 2.79 (d, *J* = 16.2 Hz, 1H), 2.43 (s, 3H), 2.24–2.13 (m, 2H), 1.73–1.61 (m, 1H); ^13^C NMR (126 MHz, CDCl_3_) δ: 166.0, 159.3, 146.7, 144.0, 131.9, 131.8, 130.9, 129.1, 129.0, 122.7, 122.4, 121.2, 119.4, 113.9, 111.3, 86.3, 63.6, 60.3, 55.8, 55.2, 53.6, 48.1, 41.7, 34.2, 27.8; ESI-HRMS *m/z* calcd for C_25_H_28_NO_5_ [M + H]^+^ 422.1962 found 422.1972.

#### 3.3.12. 6-*O*-(4-methoxybenzoyl)galanthamine (**1l**)

Yield 45 mg (63%); white amorphous solid; $\left[\alpha \right]{\;}_{D}^{28}$= −73.6° (c 0.12, MeOH); ^1^H NMR (500 MHz, CDCl_3_) δ: 8.06–7.91 (m, 2H, AA’BB‘), 6.90–6.78 (m, 2H, AA’BB‘), 6.67 (d, *J* = 8.1 Hz, 1H), 6.57 (d, *J* = 8.1 Hz, 1H), 6.34 (d, *J* = 10.4 Hz, 1H), 6.03 (dd, *J* = 10.4 Hz, *J* = 5.2 Hz, 1H), 5.58–5.49 (m, 1H), 4.67–4.62 (m, 1H), 4.14 (d, *J* = 15.3 Hz, 1H), 3.88 (s, 3H), 3.82 (s, 3H), 3.67 (d, *J* = 15.3 Hz, 1H), 3.33 (t, *J* = 13.6 Hz, 1H), 3.11–3.00 (m, 1H), 2.81–2.67 (m, 1H), 2.40 (s, 3H), 2.25–2.01 (m, 2H), 1.68–1.56 (m, 1H); ^13^C NMR (126 MHz, CDCl_3_) δ: 165.9, 163.1, 146.7, 143.9, 132.0, 131.9, 130.8, 129.3, 123.0, 123.0, 121.1, 113.3, 111.4, 86.3, 63.2, 60.3, 56.0, 55.2, 53.6, 48.1, 41.7, 34.2, 28.0; ESI-HRMS *m/z* calcd for C_25_H_28_NO_5_ [M + H]^+^ 422.1962 found 422.1972.

#### 3.3.13. 6-*O*-(3,4-dimethoxybenzoyl)galanthamine (**1m**)

Yield 54 mg (69%); white amorphous solid; $\left[\alpha \right]{\;}_{D}^{28}$= −26.7° (c 0.07, MeOH); ^1^H NMR (500 MHz, CDCl_3_) δ: 7.71 (d, *J* = 8.6 Hz, 1H), 7.55 (s, 1H), 6.83 (d, *J* = 8.6 Hz, 1H), 6.67 (d, *J* = 8.1 Hz, 1H), 6.59 (d, *J* = 8.1 Hz, 1H), 6.33 (d, *J* = 10.3 Hz, 1H), 6.03 (dd, *J* = 10.3 Hz, *J* = 5.0 Hz, 1H), 5.66–5.49 (m, 1H), 4.63 (s, 1H), 4.15 (d, *J* = 15.1 Hz, 1H), 3.91 (s, 6H), 3.83 (s, 3H), 3.69 (d, *J* = 15.1 Hz, 1H), 3.34 (t, *J* = 13.6 Hz, 1H), 3.14–3.02 (m, 1H), 2.84–2.68 (m, 1H), 2.41 (s, 3H), 2.25–2.08 (m, 2H), 1.71–1.52 (m, 1H); ^13^C NMR (126 MHz, CDCl_3_) δ: 165.9, 152.8, 148.4, 146.7, 144.0, 132.1, 130.6, 128.9, 124.0, 123.0, 121.3, 112.1, 111.3, 110.0, 86.4, 63.3, 60.3, 55.9, 55.8, 55.7, 53.7, 48.0, 41.7, 34.2, 27.9; ESI-HRMS *m/z* calcd for C_26_H_30_NO_6_ [M + H]^+^ 452.2068 found 452.2074.

#### 3.3.14. 6-*O*-(3,5-dimethoxybenzoyl)galanthamine (**1n**)

Yield 75 mg (95%); white amorphous solid; $\left[\alpha \right]{\;}_{D}^{28}$= −32.0° (c 0.25, MeOH); ^1^H NMR (500 MHz, CDCl_3_) δ: 7.22 (d, *J* = 2.4 Hz, 2H), 6.68 (d, *J* = 8.1 Hz, 1H), 6.64–6.58 (m, 2H), 6.34 (d, *J* = 10.3 Hz, 1H), 6.03 (dd, *J* = 10.3, 5.3 Hz, 1H), 5.63–5.56 (m, 1H), 4.67–4.62 (m, 1H), 4.17 (d, *J* = 15.1 Hz, 1H), 3.83 (s, 3H), 3.82 (s, 6H), 3.72 (d, *J* = 15.1 Hz, 1H), 3.37 (t, *J* = 13.6 Hz, 1H), 3.17–3.05 (m, 1H), 2.83–2.70 (m, 1H), 2.43 (s, 3H), 2.23–2.09 (m, 2H), 1.71–1.57 (m, 1H); ^13^C NMR (126 MHz, CDCl_3_) δ: 166.0, 160.5, 146.8, 144.2, 132.4, 132.0, 130.8, 122.8, 121.4, 111.3, 107.3, 105.9, 86.3, 63.7, 60.2, 55.8, 55.4, 53.6, 48.0, 41.6, 34.1, 27.9; ESI-HRMS m/z calcd for C_26_H_30_NO_6_ [M + H]^+^ 452.2068 found 452.2075.

#### 3.3.15. 6-*O*-(3,4,5-trimethoxybenzoyl)galanthamine (**1o**)

Yield 61 mg (75%); white amorphous solid; $\left[\alpha \right]{\;}_{D}^{28}$= −10.3° (c 0.35, MeOH); ^1^H NMR (500 MHz, CDCl_3_) δ: 7.32 (s, 2H), 6.66 (d, *J* = 8.1 Hz, 1H), 6.60 (d, *J* = 8.1 Hz, 1H), 6.33 (d, *J* = 10.4 Hz, 1H), 6.01 (dd, *J* = 10.4 Hz, *J* = 5.1 Hz, 1H), 5.64–5.54 (m, 1H), 4.64–4.60 (m, 1H), 4.16 (d, *J* = 15.0 Hz, 1H), 3.89 (s, 6H), 3.87 (s, 3H), 3.79 (s, 3H), 3.71 (d, *J* = 15.0 Hz, 1H), 3.35 (t, *J* = 13.5 Hz, 1H), 3.16–3.05 (m, 1H), 2.80–2.70 (m, 1H), 2.42 (s, 3H), 2.23–2.11 (m, 2H), 1.70–1.57 (m, 1H); ^13^C NMR (126 MHz, CDCl_3_) δ: 165.7, 152.7, 146.7, 144.0, 142.0, 132.1, 130.6, 128.4, 125.4, 122.8, 121.5, 111.0, 107.0, 86.4, 63.5, 60.8, 60.2, 56.0, 55.6, 53.6, 47.9, 41.7, 34.2, 27.8; ESI-HRMS *m/z* calcd for C_27_H_32_NO_7_ [M + H]^+^ 482.2173 found 482.2174.

#### 3.3.16. 6-*O*-(3,5-diethoxybenzoyl)galanthamine (**1p**)

Yield 53 mg (64%); white amorphous solid; $\left[\alpha \right]{\;}_{D}^{28}$= −17.0° (c 0.16, MeOH); ^1^H NMR (500 MHz, CDCl_3_) δ: 7.17 (s, 2H), 6.67 (d, *J* = 8.1 Hz, 1H), 6.62–6.53 (m, 2H), 6.33 (d, *J* = 10.3 Hz, 1H), 6.00 (dd, *J* = 10.3 Hz, *J* = 5.0 Hz, 1H), 5.62–5.48 (m, 1H), 4.67–4.55 (m, 1H), 4.14 (d, *J* = 15.1 Hz, 1H), 4.02 (q, *J* = 7.0 Hz, 4H), 3.84 (s, 3H), 3.69 (d, *J* = 15.1 Hz, 1H), 3.33 (t, *J* = 13.6 Hz, 1H), 3.14–3.00 (m, 1H), 2.83–2.67 (m, 1H), 2.41 (s, 3H), 2.22–2.08 (m, 2H), 1.70–1.53 (m, 1H), 1.39 (t, *J* = 7.0 Hz, 6H); ^13^C NMR (126 MHz, CDCl_3_) δ: 166.0, 159.7, 146.8, 144.0, 132.2, 132.0, 130.8, 129.0, 122.7, 121.2, 111.6, 107.8, 106.4, 86.2, 63.7, 63.5, 60.3, 56.0, 53.7, 48.0, 41.7, 34.2, 27.8, 14.7; ESI-HRMS *m/z* calcd for C_28_H_34_NO_6_ [M + H]^+^ 480.2381 found 480.2386.

#### 3.3.17. 6-*O*-(1-naphthoyl)galanthamine (**1q**)

Yield 53 mg (70%); white amorphous solid; $\left[\alpha \right]{\;}_{D}^{28}$= −40.0° (c 0.10, MeOH); ^1^H NMR (500 MHz, CDCl_3_) δ: 8.98 (d, *J* = 8.2 Hz, 1H), 8.34 (d, *J* = 8.2 Hz, 1H), 7.99 (d, *J* = 8.2 Hz, 1H), 7.86 (d, *J* = 8.2 Hz, 1H), 7.58 (t, *J* = 8.2 Hz, 1H), 7.51 (t, *J* = 8.2 Hz, 1H), 7.45 (t, *J* = 8.2 Hz, 1H), 6.72 (d, *J* = 8.1 Hz, 1H), 6.62 (d, *J* = 8.1 Hz, 1H), 6.40 (d, *J* = 10.2 Hz, 1H), 6.12 (dd, *J* = 10.2 Hz, *J* = 5.1 Hz, 1H), 5.77–5.61 (m, 1H), 4.75–4.63 (m, 1H), 4.19 (d, *J* = 15.1 Hz, 1H), 3.89 (s, 3H), 3.73 (d, *J* = 15.1 Hz, 1H), 3.39 (t, *J* = 13.5 Hz, 1H), 3.22–3.04 (m, 1H), 2.96–2.81 (m, 1H), 2.44 (s, 3H), 2.30–2.12 (m, 2H), 1.75–1.61 (m, 1H); ^13^C NMR (126 MHz, CDCl_3_) δ: 167.1, 146.7, 144.1, 133.7, 133.2, 132.1, 131.5, 131.4, 131.0, 128.8, 128.4, 127.6, 127.1, 125.9, 125.9, 124.5, 123.0, 121.5, 111.5, 86.4, 63.7, 60.3, 56.0, 53.7, 48.1, 41.6, 34.1, 27.9; ESI-HRMS *m/z* calcd for C_28_H_28_NO_4_ [M + H]^+^ 442.2013 found 442.2020.

#### 3.3.18. 6-*O*-(2-naphthoyl)galanthamine (**1r**)

Yield 56 mg (74%); white amorphous solid; $\left[\alpha \right]{\;}_{D}^{28}$= −35.2° (c 0.25, MeOH); ^1^H NMR (500 MHz, CDCl_3_) δ: 8.63 (s, 1H), 8.12–8.05 (m, 1H), 7.95 (d, *J* = 8.3 Hz, 1H), 7.86 (d, *J* = 8.3 Hz, 1H), 7.83 (d, *J* = 8.3 Hz, 1H), 7.60–7.47 (m, 2H), 6.73 (d, *J* = 8.1 Hz, 1H), 6.62 (d, *J* = 8.1 Hz, 1H), 6.40 (d, *J* = 10.3 Hz, 1H), 6.11 (dd, *J* = 10.3 Hz, *J* = 5.2 Hz, 1H), 5.70–5.64 (m, 1H), 4.73–4.68 (m, 1H), 4.19 (d, *J* = 15.1 Hz, 1H), 3.92 (s, 3H), 3.73 (d, *J* = 15.1 Hz, 1H), 3.39 (t, *J* = 13.6 Hz, 1H), 3.17–3.07 (m, 1H), 2.91–2.80 (m, 1H), 2.44 (s, 3H), 2.28–2.12 (m, 2H), 1.72–1.62 (m, 1H); ^13^C NMR (126 MHz, CDCl_3_) δ: 166.4, 146.8, 144.2, 135.5, 132.5, 132.1, 131.5, 131.0, 129.4, 128.0, 127.8, 127.8, 127.7, 126.2, 125.6, 123.0, 121.4, 111.6, 86.4, 63.7, 60.3, 56.1, 53.7, 48.2, 41.6, 34.2, 28.0; ESI-HRMS *m/z* calcd for C_28_H_28_NO_4_[M + H]^+^ 442.2013 found 442.2021.

#### 3.3.19. 6-*O*-(2-furoyl)galanthamine (**1s**)

Yield 38 mg (57%); white amorphous solid; $\left[\alpha \right]{\;}_{D}^{28}$= −40.0° (c 0.05, MeOH); ^1^H NMR (500 MHz, CDCl_3_) δ: 7.54 (s, 1H), 7.15 (d, *J* = 3.5 Hz, 1H), 6.68 (d, *J* = 8.0 Hz, 1H), 6.58 (d, *J* = 8.0 Hz, 1H), 6.49–6.41 (m, 1H), 6.36 (d, *J* = 10.3 Hz, 1H), 5.99 (dd, *J* = 10.3 Hz, *J* = 5.1 Hz, 1H), 5.61–5.52 (m, 1H), 4.65–4.60 (m, 1H), 4.14 (d, *J* = 15.1 Hz, 1H), 3.86 (s, 3H), 3.69 (d, *J* = 15.1 Hz, 1H), 3.34 (t, *J* = 13.7 Hz, 1H), 3.12–3.00 (m, 1H), 2.83–2.70 (m, 1H), 2.41 (s, 3H), 2.22–2.07 (m, 2H), 1.67–1.54 (m, 1H); ^13^C NMR (126 MHz, CDCl_3_) δ: 158.4, 146.7, 146.3, 144.8, 144.0, 131.9, 131.3, 129.2, 122.5, 121.3, 118.1, 111.7, 111.6, 86.1, 63.7, 60.3, 56.0, 53.7, 48.1, 41.6, 34.1, 27.9; ESI-HRMS *m/z* calcd for C_22_H_24_NO_5_ [M + H]^+^ 382.1649 found 382.1658.

#### 3.3.20. 3-*O*-(1-naphthoyl)vittatine (**4a**)

Yield 51 mg (79%); pale yellow amorphous solid; $\left[\alpha \right]{\;}_{D}^{24}$ = –115.2° (c 0.12, CHCl_3_); ^1^H NMR (500 MHz, CDCl_3_) *δ*: 8.86 (d, *J* = 8.1 Hz, 1H), 8.09 (dd, *J* = 8.1 Hz, *J* = 1.0 Hz, 1H), 7.99 (d, *J* = 8.1 Hz, 1H), 7.86 (d, *J* = 8.1 Hz, 1H), 7.58–7.50 (m, 2H), 7.45 (t, *J* = 8.1 Hz, 1H), 6.89 (s, 1H), 6.80 (d, *J* = 9.8 Hz, 1H), 6.52 (s, 1H), 6.16 (dd, *J* = 9.8 Hz, *J* = 5.5 Hz, 1H), 5.91 (d, overlap, *J* = 11.7 Hz, 1H), 5.90 (d, overlap, *J* = 11.7 Hz, 1H), 5.75–5.71 (m, 1H), 4.45 (d, *J* = 16.9 Hz, 1H), 3.82 (d, *J* = 16.9 Hz, 1H), 3.51 (dd, *J* = 13.0 Hz, *J* = 4.4 Hz, 1H), 3.43 (ddd, *J* = 13.0 Hz, *J* = 9.8 Hz, *J* = 5.0 Hz, 1H), 2.96 (ddd, *J* = 13.0 Hz, *J* = 9.8 Hz, *J* = 5.0 Hz, 1H), 2.29–2.20 (m, 2H), 2.06–1.94 (m, 2H); ^13^C NMR (126 MHz, CDCl_3_) *δ*: 167.0, 146.2, 145.8, 138.2, 134.9, 133.8, 133.2, 131.3, 130.1, 128.5, 127.6, 127.4, 126.5, 126.1, 125.8, 124.4, 123.8, 107.0, 102.8, 100.8, 67.3, 63.5, 62.4, 53.6, 44.4, 44.2, 30.1; ESI-HRMS m/z calcd for C_27_H_24_NO_4_ [M + H]^+^ 427.1700 found 427.1699.

#### 3.3.21. 3-*O*-(2-naphthoyl)vittatine (**4b**)

Yield 56 mg (88%); pale yellow amorphous solid; $\left[\alpha \right]{\;}_{D}^{24}$= –75.0° (c 0.11, CHCl_3_); ^1^H NMR (500 MHz, CDCl_3_) *δ*: 8.56 (bs, 1H), 8.02 (dd, *J* = 8.4 Hz, *J* = 1.5 Hz, 1H), 7.94 (d, *J* = 8.4 Hz, 1H), 7.86 (d, *J* = 8.0 Hz, 1H), 7.84 (d, *J* = 8.0 Hz, 1H), 7.58 (t, *J* = 8.0 Hz, 1H), 7.52 (t, *J* = 8.0 Hz, 1H), 6.91 (s, 1H), 6.80 (d, *J* = 9.8 Hz, 1H), 6.56 (s, 1H), 6.11 (dd, *J* = 9.8 Hz, *J* = 5.6 Hz, 1H), 5.94–5.92 (m, 1H), 5.91–5.90 (m, 1H), 5.70–5.67 (m, 1H), 4.49 (d, *J* = 17.1 Hz, 1H), 3.85 (d, *J* = 17.1 Hz, 1H), 3.54 (dd, *J* = 12.7 Hz, *J* = 5.0 Hz, 1H), 3.44 (ddd, *J* = 12.7 Hz, *J* = 9.3 Hz, *J* = 5.0 Hz, 1H), 2.97 (ddd, *J* = 12.7 Hz, *J* = 9.3 Hz, *J* = 5.0 Hz, 1H), 2.56 (ddd, *J* = 12.7 Hz, *J* = 9.3 Hz, *J* = 5.0 Hz, 1H), 2.23–2.17 (m, 1H), 2.05–2.00 (m, 1H), 1.96 (td, *J* = 12.7 Hz, *J* = 5.0 Hz, 1H); ^13^C NMR (126 MHz, CDCl_3_) *δ*: 166.2, 146.3 145.9, 138.1, 135.5, 134.7, 132.4, 131.1, 129.3, 128.2, 128.0, 127.7, 127.6, 126.53, 126.50, 125.3, 123.8, 107.0, 102.9, 100.8, 67.3, 63.5, 62.4, 53.6, 44.4, 44.2, 30.1; ESI-HRMS m/z calcd for C_27_H_24_NO_4_ [M + H]^+^ 427.1700 found 427.1701.

#### 3.3.22. 3-*O*-(3,5-dimethylbenzoyl)maritidine (**5a**)

Yield 37 mg (75%); white amorphous solid; $\left[\alpha \right]{\;}_{D}^{24}$= −75.6° (c 0.18, CHCl_3_); ^1^H NMR (500 MHz, CDCl_3_) δ: 7.60 (s, 2H), 7.15 (s, 1H), 6.90 (s, 1H), 6.83 (d, *J* = 10.0 Hz, 1H), 6.59 (s, 1H), 6.08 (dd, *J* = 10.0 Hz, *J* = 5.1 Hz, 1H), 5.63–5.58 (m, 1H), 4.52 (d, *J* = 16.7 Hz, 1H), 3.93–3.82 (m, overlap, 1H), 3.91 (s, overlap, 3H), 3.85 (s, overlap, 3H), 3.52 (dd, *J* = 13.5 Hz, *J* = 4.3 Hz, 1H), 3.49–3.42 (m, 1H), 3.02–2.94 (m, 1H), 2.33 (s, 6H), 2.30–2.22 (m, 1H), 2.21–2.13 (m, 1H), 2.01 (ddd, *J* = 12.1 Hz, *J* = 10.3 Hz, *J* = 5.8 Hz, 1H), 1.94 (td, *J* = 13.5 Hz, *J* = 4.3 Hz, 1H); ^13^C NMR (126 MHz, CDCl_3_) δ: 166.4, 147.5, 147.4, 137.9, 136.8, 134.5, 134.3, 130.1, 127.3, 125.0, 123.9, 109.9, 105.8, 67.0, 63.5, 62.0, 56.1, 55.9, 53.5, 44.2, 43.9, 29.9, 21.1; ESI-HRMS *m/z* calcd for C_26_H_30_NO_4_ [M + H]^+^ 420.2170 found 420.2172.

#### 3.3.23. 3-*O*-(1-naphthoyl)maritidine (**5b**)

Yield 35 mg (82%); white amorphous solid; $\left[\alpha \right]{\;}_{D}^{24}$= −92.4° (c 0.19, CHCl_3_); ^1^H NMR (600 MHz, CDCl_3_) δ: 8.84 (d, *J* = 7.8 Hz, 1H), 8.07 (d, *J* = 7.8 Hz, 1H), 7.96 (d, *J* = 7.8 Hz, 1H), 7.84 (d, *J* = 7.8 Hz, 1H), 7.54–7.46 (m, 2H), 7.42 (t, *J* = 7.8 Hz, 1H), 6.88–6.83 (m, 2H), 6.54 (s, 1H), 6.16 (dd, *J* = 10.0 Hz, *J* = 5.2 Hz, 1H), 5.73–5.69 (m, 1H), 4.47 (d, *J* = 16.8 Hz, 1H), 3.93–3.77 (m, overlap, 1H), 3.87 (s, overlap, 3H), 3.81 (s, overlap, 3H), 3.53 (dd, *J* = 13.2 Hz, *J* = 4.3 Hz, 1H), 3.47–3.40 (m, 1H), 3.00–2.92 (m, 1H), 2.29–2.19 (m, 2H), 2.06–1.93 (m, 2H); ^13^C NMR (151 MHz, CDCl_3_) δ: 167.0, 147.5, 147.4, 136.9, 134.7, 133.8, 133.2, 131.3, 130.1, 128.5, 127.6, 127.3, 126.1, 125.7, 125.1, 124.4, 123.8, 110.0, 105.8, 67.3, 63.6, 62.0, 56.1, 55.9, 53.6, 44.3, 44.0, 30.0; ESI-HRMS *m/z* calcd for C_28_H_28_NO_4_ [M + H]^+^ 442.2013 found 442.2019.

### 3.4. Preparation of 3-O-methylpancracine (***3***)

3-*O*-methylpancracine (3) was synthesized through the rearrangement of haemanthamine (2), as described by Govindaraju et al. [23]. One hundred milligrams of haemanthamine was dissolved in 3 ml of dry pyridine and treated with 100 µL of methanesulfonyl chloride. After stirring at 0 °C for 8 h, the reaction mixture was poured into a solution of 100 mg of sodium bicarbonate in 10 ml of distilled water. The solution was again stirred overnight, and then extracted with chloroform, 2 times. The solution was treated with anhydrous sodium sulfate, filtered and the solvent was removed under reduced pressure. The residue was purified by preparative TLC (To:Et_2_NH 9:1), resulting in 70 mg of 3-*O*-methylpancracine (70%) as an amorphous white solid. The reaction was repeated several times during the study to provide compound 3 for further derivatization.

$\left[\alpha \right]{\;}_{D}^{21}$= –51.8° (c 0.25, CHCl_3_); ^1^H NMR (500 MHz, CDCl_3_) δ: 6.55 (s, 1H), 6.47 (s, 1H), 5.88 (d, overlap, *J* = 13.5 Hz, 1H), 5.88 (d, overlap, *J* = 13.5 Hz, 1H), 5.59–5.51 (m, 1H), 4.36 (d, *J* = 16.6 Hz, 1H), 4.10–4.04 (m, 1H), 3.84 (d, *J* = 16.6 Hz, 1H), 3.59–3.52 (m, 1H), 3.41–3.34 (m, overlap, 1H), 3.39 (s, overlap, 3H), 3.30 (d, *J* = 2.6 Hz, 1H), 3.12 (dd, *J* = 11.2, *J* = 2.6 Hz, 1H), 3.06 (d, *J* = 11.2 Hz, 1H), 2.35–2.26 (m, 1H), 1.54 (td, *J* = 12.5, *J* = 3.2 Hz, 1H); ^13^C NMR (126 MHz, CDCl_3_) δ: 153.7, 146.8, 146.0, 132.1, 124.3, 115.1, 107.4, 106.8, 100.8, 81.3, 67.3, 60.8, 58.8, 57.0, 55.5, 45.5, 28.6; ESI-HRMS *m/z* calcd for C_17_H_20_NO_4_ [M + H]^+^ 302.1378 found 302.1386. 

### 3.5. Preparation of 3-O-methylpancracine (***3a***–***3g***) Derivatives

The same procedure as that described previously was used to afford the corresponding esters **3a**–**3g** [29].

#### 3.5.1. 2-*O*-(3,5-dimethylbenzoyl)-3-*O*-methylpancracine (**3a**)

Yield 23 mg (58%); white amorphous solid; $\left[\alpha \right]{\;}_{D}^{29}$= +114.3° (c 0.10, CHCl_3_); ^1^H NMR (600 MHz, CDCl_3_) δ: 7.62 (dd, *J* = 1.8 Hz, *J* = 0.7 Hz, 2H), 7.21–7.17 (m, 1H), 6.54 (s, 1H), 6.46 (s, 1H), 5.87 (d, overlap, *J* = 13.8 Hz, 1H), 5.87 (d, overlap, *J* = 13.8 Hz, 1H), 5.56–5.53 (m, 1H), 5.39–5.35 (m, 1H), 4.36 (d, *J* = 16.6 Hz, 1H), 3.85 (d, *J* = 16.6 Hz, 1H), 3.69–3.64 (m, 1H), 3.46–3.42 (m, overlap, 1H), 3.44 (s, overlap, 3H), 3.31 (d, *J* = 2.3 Hz, 1H), 3.13 (dd, *J* = 11.3 Hz, *J* = 2.3 Hz, 1H), 3.07 (d, *J* = 11.3 Hz, 1H), 2.39–2.32 (m, overlap, 1H), 2.36 (s, overlap, 6H), 1.56 (td, *J* = 12.4 Hz, *J* = 2.7 Hz, 1H); ^13^C NMR (151 MHz, CDCl_3_) δ: 166.0, 155.4, 146.9, 146.1, 138.1, 134.8, 131.7, 129.9, 127.3, 124.2, 111.6, 107.5, 106.8, 100.8, 78.4, 68.3, 60.9, 58.4, 57.3, 55.5, 45.7, 30.0, 21.1; ESI-HRMS m/z calcd for C_26_H_28_NO_5_ [M + H]^+^ 434.1962 found 434.1965.

#### 3.5.2. 2-*O*-(2,4,6-trimethylbenzoyl)-3-*O*-methylpancracine (**3b**)

Yield 29 mg (42%); white amorphous solid; $\left[\alpha \right]{\;}_{D}^{28}$= +17.6° (c 0.41, CHCl_3_); ^1^H NMR (500 MHz, CDCl_3_) δ: 6.87 (s, 2H), 6.56 (s, 1H), 6.48 (s, 1H), 5.89 (d, overlap, *J* = 11.4 Hz, 1H), 5.89 (d, overlap, *J* = 11.4 Hz, 1H), 5.60–5.57 (m, 1H), 5.46–5.41 (m, 1H), 4.36 (d, *J* = 16.6 Hz, 1H), 3.86 (d, *J* = 16.6 Hz, 1H), 3.73–3.67 (m, 1H), 3.53–3.42 (m, overlap, 1H), 3.49 (s, overlap, 3H), 3.33–3.30 (m, 1H), 3.10–3.03 (m, 2H), 2.40–2.33 (m, 1H), 2.32 (s, 6H), 2.29 (s, 3H), 1.53 (td, *J* = 12.3 Hz, *J* = 2.7 Hz, 1H); ^13^C NMR (126 MHz, CDCl_3_) δ: 169.0, 156.0, 146.9, 146.0, 139.5, 135.0, 131.8, 130.6, 128.5, 124.5, 111.1, 107.5, 106.8, 100.8, 78.5, 68.6, 61.0, 58.2, 57.3, 55.5, 45.8, 29.9, 21.1, 19.8; ESI-HRMS m/z calcd for =C_27_H_30_NO_5_ [M + H]^+^ 448.2118 found 448.2119.

#### 3.5.3. 2-*O*-(4-tert-butylbenzoyl)-3-*O*-methylpancracine (**3c**)

Yield 28 mg (98%); white amorphous solid; $\left[\alpha \right]{\;}_{D}^{29}$= +40.9° (c 0.90, CHCl_3_); ^1^H NMR (500 MHz, CDCl_3_) δ: 8.01–7.95 (m, 2H, AA′BB′), 7.50–7.44 (m, 2H, AA′BB′), 6.55 (s, 1H), 6.48 (s, 1H), 5.89 (d, overlap, *J* = 13.4 Hz, 1H), 5.89 (d, overlap, *J* = 13.4 Hz, 1H), 5.59–5.54 (m, 1H), 5.46–5.41 (m, 1H), 4.37 (d, *J* = 16.7 Hz, 1H), 3.87 (d, *J* = 16.7 Hz, 1H), 3.71–3.64 (m, 1H), 3.50–3.42 (m, overlap, 1H), 3.46 (s, overlap, 3H), 3.32 (d, *J* = 2.1 Hz, 1H), 3.13 (dd, *J* = 11.3 Hz, *J* = 2.1 Hz, 1H), 3.09 (d, *J* = 11.3 Hz, 1H), 2.40–2.32 (m, 1H), 1.56 (td, *J* = 12.6 Hz, *J* = 2.9 Hz, 1H), 1.35 (s, 9H); ^13^C NMR (126 MHz, CDCl_3_) δ: 165.7, 156.8, 156.1, 146.8, 146.0, 132.0, 129.5, 127.3, 125.3, 124.7, 111.3, 107.5, 106.8, 100.7, 78.6, 68.3, 61.1, 58.1, 57.3, 55.5, 45.8, 35.1, 31.1, 30.2; ESI-HRMS m/z calcd for C_28_H_32_NO_5_ [M + H]^+^ 462.2275 found 462.2279.

#### 3.5.4. 2-*O*-(4-butylbenzoyl)-3-*O*-methylpancracine (**3d**)

Yield 32 mg (98%); white amorphous solid; $\left[\alpha \right]{\;}_{D}^{29}$= +48.9° (c 0.15, CHCl_3_); ^1^H NMR (500 MHz, CDCl_3_) δ: 7.98–7.92 (m, 2H, AA′BB′), 7.29–7.23 (m, 2H, AA′BB′), 6.55 (s, 1H), 6.48 (s, 1H), 5.89 (d, overlap, *J* = 13.4 Hz, 1H), 5.89 (d, overlap, *J* = 13.4 Hz, 1H), 5.59–5.54 (m, 1H), 5.42–5.37 (m, 1H), 4.36 (d, *J* = 16.7 Hz, 1H), 3.86 (d, *J* = 16.7 Hz, 1H), 3.71–3.66 (m, 1H), 3.49–3.42 (m, overlap, 1H), 3.46 (s, overlap, 3H), 3.32 (d, *J* = 2.0 Hz, 1H), 3.11 (dd, *J* = 11.1 Hz, *J* = 2.0 Hz, 1H), 3.08 (d, *J* = 11.1 Hz, 1H), 2.67 (t, *J* = 8.1 Hz, 2H), 2.39–2.31 (m, 1H), 1.66–1.58 (m, 2H), 1.55 (td, *J* = 12.0 Hz, *J* = 2.6 Hz, 1H), 1.36 (dq, *J* = 14.7 Hz, *J* = 7.3 Hz, 2H), 0.93 (t, *J* = 7.3 Hz, 3H); ^13^C NMR (126 MHz, CDCl_3_) δ: 165.8, 156.2, 148.8, 146.8, 145.9, 132.0, 129.7, 128.5, 127.5, 124.7, 111.3, 107.5, 106.8, 100.7, 78.6, 68.3, 61.1, 58.1, 57.3, 55.5, 45.8, 35.7, 33.3, 30.2, 22.2, 13.9; ESI-HRMS m/z calcd for C_28_H_32_NO_5_ [M + H]^+^ 462.2275 found 462.2281.

#### 3.5.5. 2-*O*-(3,5-diethoxybenzoyl)-3-*O*-methylpancracine (**3e**)

Yield 20 mg (48%); white amorphous solid; $\left[\alpha \right]{\;}_{D}^{29}$= +69.0**°** (c 0.14, CHCl_3_); ^1^H NMR (500 MHz, CDCl_3_) δ: 7.16 (d, *J* = 2.3 Hz, 2H), 6.65 (t, *J* = 2.3 Hz, 1H), 6.55 (s, 1H), 6.48 (s, 1H), 5.89 (d, overlap, *J* = 13.4 Hz, 1H), 5.89 (d, overlap, *J* = 13.4 Hz, 1H), 5.57–5.52 (m, 1H), 5.40–5.35 (m, 1H), 4.37 (d, *J* = 16.6 Hz, 1H), 4.06 (q, *J* = 7.0 Hz, 4H), 3.86 (d, *J* = 16.6 Hz, 1H), 3.70–3.65 (m, 1H), 3.49–3.41 (m, overlap, 1H), 3.46 (s, overlap, 3H), 3.32 (d, *J* = 2.6 Hz, 1H), 3.12 (dd, *J* = 11.3 Hz, *J* = 2.6 Hz, 1H), 3.09 (d, *J* = 11.3 Hz, 1H), 2.39–2.31 (m, 1H), 1.55 (td, *J* = 12.6 Hz, *J* = 2.4 Hz, 1H), 1.43 (t, *J* = 7.0 Hz, 6H); ^13^C NMR (126 MHz, CDCl_3_) δ: 165.6, 159.9, 156.3, 146.8, 145.9, 132.0, 131.9, 124.7, 111.1, 107.9, 107.5, 106.8, 106.2, 100.7, 78.5, 68.7, 63.8, 61.1, 58.1, 57.3, 55.5, 45.8, 30.2, 14.7; ESI-HRMS m/z calcd for C_28_H_32_NO_7_ [M + H]^+^ 494.2173 found 494.2179.

#### 3.5.6. 2-*O*-(1-naphthoyl)-3-*O*-methylpancracine (**3f**)

Yield 38 mg (100%); white amorphous solid; $\left[\alpha \right]{\;}_{D}^{28}$= +30.8° (c 0.19, CHCl_3_); ^1^H NMR (500 MHz, CDCl_3_) δ: 8.93 (d, *J* = 8.0 Hz, 1H), 8.18 (d, *J* = 8.0 Hz, 1H), 8.04 (d, *J* = 8.0 Hz, 1H), 7.90 (d, *J* = 8.0 Hz, 1H), 7.64 (t, *J* = 8.0 Hz, 1H), 7.59–7.48 (m, 2H), 6.58 (s, 1H), 6.49 (s, 1H), 5.90 (d, overlap, *J* = 12.4 Hz, 1H), 5.90 (d, overlap, *J* = 12.4 Hz, 1H), 5.68–5.65 (m, 1H), 5.56–5.51 (m, 1H), 4.38 (d, *J* = 16.6 Hz, 1H), 3.88 (d, *J* = 16.6 Hz, 1H), 3.81–3.78 (m, 1H), 3.59–3.46 (m, overlap, 1H), 3.53 (s, overlap, 3H), 3.35 (d, *J* = 2.1 Hz, 1H), 3.14 (dd, *J* = 11.2 Hz, *J* = 2.1 Hz, 1H), 3.09 (d, *J* = 11.2 Hz, 1H), 2.45–2.37 (m, 1H), 1.62 (td, *J* = 13.0 Hz, *J* = 2.4 Hz, 1H); ^13^C NMR (126 MHz, CDCl_3_) δ: 166.5, 156.4, 146.8, 145.9, 133.8, 133.5, 132.0, 131.3, 130.2, 128.6, 127.8, 127.0, 126.2, 125.6, 124.7, 124.4, 111.2, 107.5, 106.8, 100.7, 78.6, 68.7, 61.1, 58.1, 57.3, 55.6, 45.8, 30.2; ESI-HRMS m/z calcd for C_28_H_26_NO_5_ [M + H]^+^ 456.1805 found 456.1800.

#### 3.5.7. 2-*O*-(2-naphthoyl)-3-*O*-methylpancracine (**3g**)

Yield 39 mg (100%); white amorphous solid; $\left[\alpha \right]{\;}_{D}^{28}$= +86.0° (c 0.15, CHCl_3_); ^1^H NMR (500 MHz, CDCl_3_) δ: 8.61 (d, *J* = 1.7 Hz, 1H), 8.06 (dd, *J* = 8.4 Hz, *J* = 1.7 Hz, 1H), 7.98 (d, *J* = 8.4 Hz, 1H), 7.90 (d, *J* = 8.4 Hz, 2H), 7.65–7.53 (m, 2H), 6.57 (s, 1H), 6.49 (s, 1H), 5.93–5.84 (m, 2H), 5.66–5.61 (m, 1H), 5.51–5.46 (m, 1H), 4.38 (d, *J* = 16.6 Hz, 1H), 3.88 (d, *J* = 16.6 Hz, 1H), 3.79–3.74 (m, 1H), 3.53–3.45 (m, overlap, 1H), 3.50 (s, overlap, 3H), 3.36 (d, *J* = 2.5 Hz, 1H), 3.17 (dd, *J* = 11.1 Hz, *J* = 2.5 Hz, 1H), 3.11 (d, *J* = 11.1 Hz, 1H), 2.44–2.36 (m, 1H), 1.63 (td, *J* = 12.4, *J* = 2.7 Hz, 1H);^13^C NMR (126 MHz, CDCl_3_) δ: 165.9, 156.4, 146.8, 146.0, 135.6, 132.5, 132.0, 131.1, 129.3, 128.4, 128.2, 127.8, 127.4, 126.7, 125.2, 124.8, 111.2, 107.5, 106.8, 100.7, 78.6, 68.8, 61.2, 58.2, 57.3, 55.6, 45.8, 30.3; ESI-HRMS m/z calcd for C_28_H_26_NO_5_ [M + H]^+^ 456.1805 found 456.1808.

### 3.6. Conversion of Derivatives to Hydrochlorides

The synthesized derivatives were dissolved in ethereal HCl and converted to the corresponding hydrochlorides.

### 3.7. Antimycobacterial Screening

The antimycobacterial assay was performed with fast-growing *Mycobacterium smegmatis* DSM 43465 (ATCC 607), *Mycobacterium aurum* DSM 43999 (ATCC 23366) from the German Collection of Microorganisms and Cell Cultures (Braunschweig, Germany) and an avirulent strain of *Mycobacterium tuberculosis* H37Ra ITM-M006710 (ATCC 9431) from Belgian Co-ordinated Collections of Micro-organisms (Antwerp, Belgium). The technique used for activity determination was the microdilution broth panel method using 96-well microtitration plates. The culture medium was Middlebrook 7H9 broth (MB) (Sigma-Aldrich, Steinheim, Germany) enriched with 0.4% glycerol (Sigma-Aldrich, Steinheim, Germany) and 10% Middlebrook OADC growth supplement (Himedia, Mumbai, India).

The mycobacterial strains were cultured on Middlebrook 7H9 agar and suspensions were prepared in Middlebrook 7H9 broth. The final density was adjusted to 1.0 on the McFarland scale and diluted in z ratio of either 1:20 (for fast-growing mycobacteria) or 1:10 (for *M. tuberculosis*) with broth.

The test compounds were dissolved in DMSO (Sigma-Aldrich, Steinheim, Germany), then MB broth was added to obtain a concentration of 2000 µg/mL. The standards used for activity determination were isoniazid (INH), rifampicin (RIF) and ciprofloxacin (CPX) (Sigma-Aldrich, Steinheim, Germany). Final concentrations were reached by binary dilution and the addition of mycobacterial suspension, and were set as 500, 250, 125, 62.5, 31.25, 15.625, 7.81 and 3.91 µg/mL. Isoniazid was diluted in the range of 500-3.91 µg/mL for screening against fast-growing mycobacteria and 1-0.0078 µg/mL for screening against *M. tuberculosis*. Rifampicin final concentrations ranged from 50 to 0.39 µg/mL for fast-growing mycobacteria and from 1 to 0.0078 µg/mL for *M. tuberculosis*. Ciprofloxacin was used for screening antimycobacterial activity with final concentrations of 1, 0.5, 0.25, 0.125, 0.0625, 0.0313, 0.0156 and 0.0078 µg/mL. The final concentration of DMSO did not exceed 2.5% (*v/v*) and did not affect the growth of *M. smegmatis*, *M. aurum* and *M. tuberculosis*. Positive (broth, DMSO, bacteria) and negative (broth, DMSO) growth controls were included.

Plates were sealed with polyester adhesive film and incubated in the dark at 37 °C without agitation. The addition of a 0.01% solution of resazurin sodium salt followed after 48 h of incubation for *M. smegmatis*, after 72 h for *M. aurum* and after 120 h for *M. tuberculosis*. Microtitration panels were then incubated for a further 2.5 h for the determination of activity against *M. smegmatis*, 4 hours for *M. aurum* and 18 hours for *M. tuberculosis*.

The antimycobacterial activity was expressed as the minimal inhibition concentration (MIC) and the value was read on the basis of stain color change (blue color—active compound; pink color—inactive compound). MIC values for standards were in the range of 7.81–15.625 µg/mL for INH, 12.5–25 µg/mL for RIF and 0.0625–0.125 µg/mL for CPX against *M. smegmatis*, 1.95–3.91 µg/mL for INH, 0.39–0.78 µg/mL for RIF and 0.0078–0.0156 µg/mL for CPX against *M. aurum* and 0.125–0.25 µg/mL for INH, 0.0039–0.0078 µg/mL for RIF and 0.125–0.25 for CPX against *M. tuberculosis*. All experiments were conducted in duplicate.

### 3.8. Cytotoxicity Assay

Human hepatocellular carcinoma HepG2 cells (ATCC HB-8065; passage 20–25) purchased from Health Protection Agency Culture Collections (ECACC, Salisbury, UK) were cultured in the Minimum Essential Eagle Medium supplemented with 10% fetal bovine serum and 1% l-glutamine solution (Sigma-Aldrich, St. Louis, MO, USA) at 37 °C in a humidified atmosphere containing 5% CO_2_. For passaging, the cells were treated with trypsin/EDTA (Sigma-Aldrich, St. Louis, MO, USA) at 37 °C and then harvested. For the cytotoxicity evaluation, the cells treated with the test substances were used while untreated HepG2 cells served as the control group; no cell control was included. The cells were seeded in a 96-well plate at a density of 50,000 cells per well and incubated for 24 h. The following day, the cells were treated with each of the test compounds dissolved in DMSO (the highest DMSO concentration used was 0.5% *v/v*). The test substances were prepared at different concentrations in triplicate according to their solubility and incubated for 24 h in a humidified atmosphere containing 5% CO_2_ at 37 °C. After incubation, a solution of thiazolyl blue tetrazolium bromide (Sigma-Aldrich, St. Louis, MO, USA) in RPMI 1640 without phenol red (BioTech) was added and incubated for 30 min in a humidified atmosphere containing 5% CO_2_ at 37 °C. Then, the formazan crystals formed were dissolved in DMSO and the absorbance of the samples was recorded at 570 nm (BioTek, Synergy Neo2 Multi-Mode Reader NEO2SMALPHAB). IC_50_ values were calculated by nonlinear regression from a semi-logarithmic plot of incubation concentration versus the percentage of absorbance relative to untreated controls using GraphPad Prism software (version 9; GraphPad Software, Inc., La Jolla, CA, USA). The obtained results of the experiments are presented as the concentration that reduces the viability of cells from the maximal viability to 50% (IC_50_).

## 4. Conclusions

Amaryllidaceae alkaloids and some of their derivatives have demonstrated a wide range of biological activities, including antimicrobial potential. On the other hand, no information about the antimycobacterial activity of AAs was available in the literature, thus the selected AAs, previously isolated from various Amaryllidaceae plants, have been initially screened for antimycobacterial activity. Moreover, pilot semisynthetic derivatives of galanthamine, 3-*O*-methyl pancracine, vittatine and maritidine have been synthesized and tested. Interestingly, the natural AAs lacked any antimycobacterial activity, but on the other hand, several semisynthetic derivatives showed great antimycobacterial potential. The most active derivatives were also evaluated for their cytotoxic activity on HepG2 cells, and thus the SI was also determined. Acylchlorides (namely 4-*tert*-butylbenzoyl chloride, 4-butylbenzoyl chloride, 3,5-diethoxybenzoyl chloride, 1-naphthoylchloride and 2-naphthoylchloride), which were used for the preparation of active derivatives in this study, will be used for the preparation of esters of other AAs previously isolated in our laboratory (e.g., haemanthamine, lycorine, galanthine) in order to study the effect of the structural type of AAs on antimycobacterial and cytotoxic activity. After additional isolation of vittatine, maritidine and other AAs, this pilot study will continue with the further structure modification of selected alkaloids. The second series of galanthamine derivatives, especially aromatic ethers and amines, will be synthesized for more detailed SAR studies. Based on the obtained results, compounds **1i** and **1r** were selected for further structure optimalization to improve the antimycobacterial potential and selectivity index by reducing the cytotoxicity. The selected compounds, showing both great antitubercular activity and low cytotoxicity, will be further subjected to searching of partner compounds from a portfolio of anti-TB drugs by checkerboard studies, and an evaluation of in vivo toxicity and in vivo treatment assays by the establishment of the *Galleria mellonella* animal model. This pilot study indicated the potential of semisynthetic derivatives of AAs for further structure optimalization as promising antimycobacterial compounds.

## Figures and Tables

**Figure 1 molecules-26-06023-f001:**
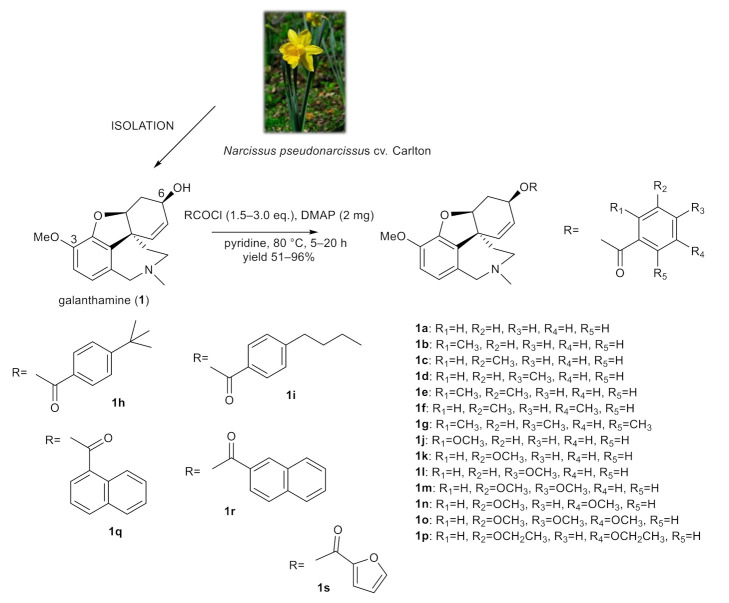
Chemical procedures affording galanthamine derivatives (**1a**–**1s**).

**Figure 2 molecules-26-06023-f002:**
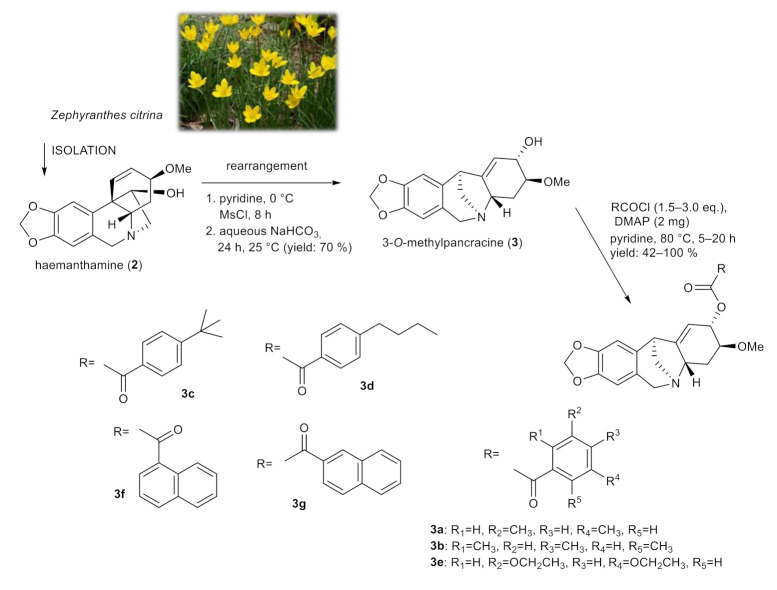
Chemical procedures affording 3-*O*-methylpancracine derivatives (**3a**–**3g**).

**Figure 3 molecules-26-06023-f003:**
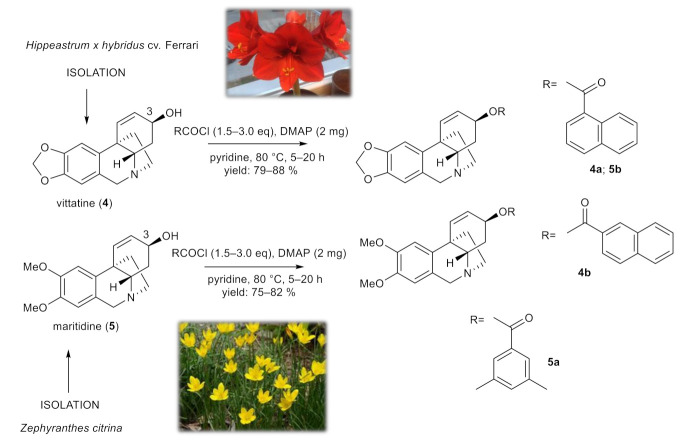
Chemical procedures affording vittatine (**4a**,**4b**) and maritidine derivatives (**5a,5b**).

**Figure 4 molecules-26-06023-f004:**
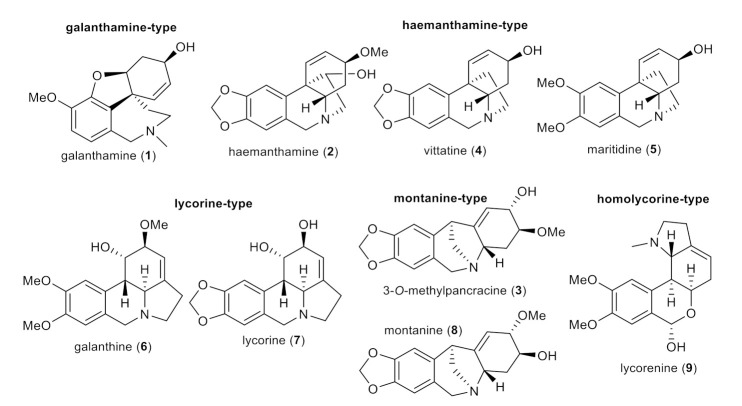
Structures of tested Amaryllidaceae alkaloids of different structural types.

**Table 1 molecules-26-06023-t001:** In vitro antimycobacterial activity against *Mtb* H37Ra, *Mycobacterium aurum* and *M. smegmatis* (MIC), cytotoxicity (IC_50_), selectivity index (SI) and calculated lipophilicity (logP, ClogP) of Amaryllidaceae alkaloids and their derivatives.

Alkaloid/Derivative	Mtb H37Ra (µg/mL)	Mtb H37Ra (µM) ^a^	*M. smegmatis* (µg/mL)	*M. aurum* (µg/mL)	HepG2 IC_50_ (µM)	SI ^b^	log*P* ^c^	ClogP ^c^
galanthamine (**1**)	≥500	≥1740.0	≥500	≥500	n.s.	n.c.	1.41	1.03
haemanthamine (**2**)	≥500	≥1659.3	≥500	≥500	n.s.	n.c.	0.98	0.37
3-*O*-methylpancracine (**3**)	≥500	≥1659.3	≥500	≥500	n.s.	n.c.	0.62	0.61
vittatine (**4**)	≥500	≥1842.9	≥500	250	n.s.	n.c.	1.44	1.08
maritidine (**5**)	≥500	≥1740.0	≥500	≥500	n.s.	n.c.	1.41	0.78
galanthine (**6**)	250	787.6	≥500	250	n.s.	n.c.	0.68	0.85
lycorine (**7**)	≥500	≥1740.2	≥500	≥500	n.s.	n.c.	0.35	0.39
montanine (**8**)	≥500	≥1659.3	≥500	≥500	n.s.	n.c.	0.62	0.61
lycorenine (**9**)	≥125	≥393.8	≥125	≥125	n.s.	n.c.	1.42	0.92
**1a**	31.25	79.8	≥500	31.25	n.s.	n.c.	3.53	3.90
**1b**	15.625	35.4	31.25	31.25	n.s.	n.c.	4.02	4.40
**1c**	15.625	35.4	31.25	15.625	n.s.	n.c.	4.02	4.40
**1d**	15.625	35.4	31.25	15.625	n.s.	n.c.	4.02	4.40
**1e**	7.81	17.1	31.25	15.625	17.9 ± 2.6	1.05	4.51	4.85
**1f**	7.81	17.1	7.81	7.81	17.6 ± 4.7	1.03	4.51	4.90
**1g**	7.81	18.0	31.25	15.625	13.9 ± 0.8	0.77	5.00	5.40
**1h**	3.125	6.9	7.81	7.81	16.8 ± 4.2	2.43	5.24	5.73
**1i**	1.56	3.5	7.81	1.98	14.7 ± 1.6	4.20	5.27	5.99
**1j**	62.5	148.3	250	125	n.s.	n.c.	3.41	3.83
**1k**	31.25	68.3	62.5	31.25	n.s.	n.c.	3.41	4.07
**1l**	31.25	68.3	62.5	31.25	n.s.	n.c.	3.41	4.07
**1m**	62.5	128.2	250	125	n.s.	n.c.	3.28	3.78
**1n**	15.625	34.6	31.25	31.25	n.s.	n.c.	3.28	4.13
**1o**	62.5	120.7	250	125	n.s.	n.c.	3.15	3.40
**1p**	3.91	7.6	15.625	7.81	20.1 ± 2.1	2.64	3.96	5.19
**1q**	6.25	13.1	7.81	7.81	42.1 ± 4.1	3.21	4.53	5.07
**1r**	1.98	4.1	7.81	3.91	21.2 ± 3.8	5.17	4.53	5.07
**1s**	62.5	149.6	≥500	250	n.s.	n.c.	2.15	3.08
**3a**	7.81	16.6	15.625	7.81	40.6 ± 7.6	2.45	3.72	4.20
**3b**	7.81	16.1	15.625	7.81	31.2 ± 5.4	1.94	4.21	4.70
**3c**	3.91	7.9	7.81	7.81	24.3 ± 2.4	3.08	4.45	5.03
**3d**	3.91	7.9	7.81	7.81	20.5 ± 0.7	2.59	4.49	5.29
**3e**	3.91	7.8	15.625	7.81	24.9 ± 1.7	3.15	3.17	4.49
**3f**	3.91	7.9	3.91	3.91	27.7 ± 4.7	3.51	3.75	4.38
**3g**	15.625	31.8	15.625	15.625	n.s.	n.c.	3.75	4.38
**4a**	3.91	8.5	7.81	3.91	23.7 ± 2.2	2.79	4.56	5.13
**4b**	3.91	8.5	7.81	3.91	18.5 ± 2.0	2.17	4.56	5.13
**5a**	3.91	8.6	15.625	15.625	18.7 ± 2.7	2.17	4.51	4.65
**5b**	7.81	16.3	15.625	7.81	16.8 ± 7.1	1.03	4.53	4.82
isoniazid ^d^	0.25	1.82	31.25	3.91	n.s.	n.c.	−0.64	−0.67
rifampicin ^d^	0.00625	0.0075	6.25	0.39	n.s.	n.c.	2.70	3.71
ciprofloxacin ^d^	0.25	0.75	0.125	0.015625	n.s.	n.c.	1.32	−0.62

^a^ Calculated from MIC (µM/ml), ^b^ SI—Selectivity index, values calculated from MIC against *Mtb* H37Ra as IC_50_ HepG2/MIC (in µM), ^c^ Log*P* and CLogP calculated in ChemDraw v18.1.; ^d^ standard; n.s. stands for not studied; n.c. stands for not calculated.

## Data Availability

The data presented in this study are available within the article or Appendix A.

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
