# Peer review of "Semisynthetic Derivatives of Selected Amaryllidaceae Alkaloids as a New Class of Antimycobacterial Agents"

_molecules, 2021, doi:10.3390/molecules26196023_

Round 1

Reviewer 1 Report

I recommend that authors reduce the methodology of antimycobacterial and cytotoxicity assays.
In the results section better explain the result on the SI values ​​of each compound, because it is not clear.

It is necessary to determine the toxicity in in vivo models of this type of compounds

Author Response

Dear Editor,

We have carefully revised our manuscript titled “Semisynthetic derivatives of selected Amaryllidaceae alkaloids as a new class of antimycobacterial agents " - Manuscript ID: molecules-1392146, according to the reviewers’ suggestions. All changes are highlighted in red.

Comments from the editors and reviewers:

Reviewer 1

I recommend that authors reduce the methodology of antimycobacterial and cytotoxicity assays.
In the results section better explain the result on the SI values ​​of each compound, because it is not clear.

We would like to keep the current detailed description of the methodology so that we can refer to this article in our subsequent manuscripts. Thank you for understanding.

We improved discussion for selectivity index.

It is necessary to determine the toxicity in in vivo models of this type of compounds.

As stated in conclusion, the next step will be evaluation of in vivo toxicity using Galleria mellonella animal model. This assay is currently being implemented at our workplace and the first results are expected over the next year. The selected active compounds from this and further series of derivatives of Amaryllidaceae derivatives will be evaluated soon.

In light of these changes, we are positive that our revised manuscript meets the criteria to be published in Molecules and would be of interest for all the readers from the scientific community with particular emphasis on alkaloids and drug development against neurodegenerative disorders.

Lucie Cahlikova

Reviewer 2 Report

The paper by Maafi et al. describes a series of semisynthetic Amaryllidaceae alkaloid derivatives and their potential as antimycobacterial agents. The synthesized compounds display low to moderate antimycobacterial activities, along with low SI. Despite these liabilities, they are worthy of interest as they might be a useful starting point for further optimization of their biological profile.   

The manuscript is clear and well organized. However, there are some criticisms which should be addressed before its publication:

  • Even though these compounds are interesting starting points for further optimization studies, these results are very preliminary and antimycobacterial activities and cytotoxicity need to be improved. The authors should state this more clearly in the results and discussion section.
  • The next most interesting activity, line 193: this sentence is not clear, please rephrase it
  • Please include reagents, conditions and yields in the synthetic schemes.
  • All screened AAs lacked any such activity, line 169: please rephrase this sentence.
  • The most active acylchlorides, line 257: please rephrase this sentence, as the acylchlorides were not tested for their antimycobacterial activities.

Author Response

Dear Editor,

We have carefully revised our manuscript titled “Semisynthetic derivatives of selected Amaryllidaceae alkaloids as a new class of antimycobacterial agents " - Manuscript ID: molecules-1392146, according to the reviewers’ suggestions. All changes are highlighted in red.

Comments from the editors and reviewers:

Reviewer 2

The paper by Maafi et al. describes a series of semisynthetic Amaryllidaceae alkaloid derivatives and their potential as antimycobacterial agents. The synthesized compounds display low to moderate antimycobacterial activities, along with low SI. Despite these liabilities, they are worthy of interest as they might be a useful starting point for further optimization of their biological profile.   

The manuscript is clear and well organized. However, there are some criticisms which should be addressed before its publication:

Even though these compounds are interesting starting points for further optimization studies, these results are very preliminary and antimycobacterial activities and cytotoxicity need to be improved. The authors should state this more clearly in the results and discussion section.

We modified and improved final conclusions to be more clear that these are pilot results that will lead to further optimalization of the structure.

The next most interesting activity, line 193: this sentence is not clear, please rephrase it

The sentence has been slightly modified.

Please include reagents, conditions and yields in the synthetic schemes.

Included.

All screened AAs lacked any such activity, line 169: please rephrase this sentence.

The sentence has been slightly modified.

The most active acylchlorides, line 257: please rephrase this sentence, as the acylchlorides were not tested for their antimycobacterial activities.

The sentence has been modified to be more clear.

In light of these changes, we are positive that our revised manuscript meets the criteria to be published in Molecules and would be of interest for all the readers from the scientific community with particular emphasis on alkaloids and drug development against neurodegenerative disorders.

Lucie Cahlikova

Reviewer 3 Report

Well written article. A large number of compounds - both natural and semi-synthetic, have been tested against antimycobacterial agents. It is worth conducting further research on the possibility of using the presented compounds in medicine.

Author Response

Dear Editor,

We have carefully revised our manuscript titled “Semisynthetic derivatives of selected Amaryllidaceae alkaloids as a new class of antimycobacterial agents " - Manuscript ID: molecules-1392146, according to the reviewers’ suggestions. All changes are highlighted in red.

Comments from the editors and reviewers:

Reviewer 3

Well written article. A large number of compounds - both natural and semi-synthetic, have been tested against antimycobacterial agents. It is worth conducting further research on the possibility of using the presented compounds in medicine.

Thank you very much for this evaluation of our work which will continue with detailed biological studies and further optimalization of structure.

In light of these changes, we are positive that our revised manuscript meets the criteria to be published in Molecules and would be of interest for all the readers from the scientific community with particular emphasis on alkaloids and drug development against neurodegenerative disorders.

Lucie Cahlikova